# The Potential of NORAD–PUMILIO–*RALGAPB* Regulatory Axis as a Biomarker in Breast Cancer

**DOI:** 10.3390/ncrna8060076

**Published:** 2022-11-10

**Authors:** Cristiane Sato Mara Muller, Igor Samesima Giner, Érika Pereira Zambalde, Tamyres Mingorance Carvalho, Enilze Maria de Souza Fonseca Ribeiro, Jaqueline Carvalho de Oliveira, Carolina Mathias, Daniela Fiori Gradia

**Affiliations:** 1Laboratory of Human Cytogenetics and Oncogenetics, Postgraduate Program in Genetics, Department of Genetics, Federal University of Paraná (UFPR), Curitiba 81531-980, Brazil; 2Multidisciplinary Laboratory of Food and Health (LabMAS), School of Applied Sciences (FCA), University of Campinas (UNICAMP), Limeira 13484-350, Brazil; 3Laboratory for Applied Science and Technology in Health, Carlos Chagas Institute, Oswaldo Cruz Foundation (FIOCRUZ), Curitiba 81310-020, Brazil

**Keywords:** LncRNA, NORAD, PUMILIO, *RALGAPB*, breast cancer, biomarker, regulatory axis

## Abstract

**Introduction**: Long non-coding RNAs (LncRNA) represent a heterogeneous family of RNAs that have emerged as regulators of various biological processes through their association with proteins in ribonucleoproteins complexes. The dynamic of these interactions can affect cell metabolism, including cancer development. Annually, breast cancer causes thousands of deaths worldwide, and searching for new biomarkers is pivotal for better diagnosis and treatment. **Methods**: Based on in silico prediction analysis, we focus on LncRNAs that have binding sites for PUMILIO, an RBP family involved in post-transcriptional regulation and associated with cancer progression. We compared the expression levels of these LncRNAs in breast cancer and non-tumor samples from the TCGA database. We analyzed the impact of overall and disease-free survival associated with the expression of the LncRNAs and co-expressed genes and targets of PUMILIO proteins. **Results**: Our results found NORAD as the most relevant LncRNA with a PUMILIO binding site in breast cancer, differently expressed between Luminal A and Basal subtypes. Additionally, NORAD was co-expressed in a Basal-like subtype (0.55) with the *RALGAPB* gene, a target gene of PUMILIO related to chromosome stability during cell division. **Conclusion**: These data suggest that this molecular axis may provide insights for developing novel therapeutic strategies for breast cancer.

## 1. Introduction

Except for skin cancers, breast cancer (BRCA) is the most common malignant disease among women. It is a heterogeneous disease that presents different pathological characteristics, and patients show distinct responses to treatments with varied survival along its course [1]. The molecular classification of BRCA is based on gene expression profiles for the stratification of patients, classifying them into four molecular subtypes: Luminal A and B, Her2-enriched, and Basal-like [1]. Luminal A (Lum A) patients show low-grade tumors and a good prognosis compared to Luminal B (Lum B), who have tumors with higher proliferation rates. Her2-enriched and Basal-like subtypes comprise patients with worse prognoses and higher-grade tumors [2]. The therapeutic management of breast cancer includes surgery, radiotherapy, and chemotherapy [2]. In addition, luminal subtypes respond to hormonal therapies, such as Tamoxifen, which benefits estrogen-positive patients [3]. Early-stage Her2-enriched patients may have a good outcome from the addition of trastuzumab along with chemotherapy, reducing the risk of relapse and death [4]. Lastly, patients with a mutated BRCA gene may benefit from the administration of PARP inhibitors, associated with chemotherapy [5]. Due to this great diversity, the discovery of additional molecular markers is needed to understand the disease’s outcome better, providing personalized therapies that improve breast cancer’s survival rate [6].

Molecular complexes such as ribonucleoprotein particles (RNP) are key players in cell life, and these complexes’ deregulation is often associated with genetic diseases, including cancer [7]. Proteins such as DNA methyltransferases (DNMTs), heterochromatin protein 1, MOF, MSL, DDP1, Polycomb group, and Trithorax group proteins are RBPs capable of binding to lncRNAs, and their levels fluctuate significantly in tumor samples. The lncRNA HOTAIR, one of the first known, is related to cancer, including breast cancer and hepatocellular carcinoma, and participates in chromatin modification complexes by interaction with the polycomb group protein PRC2, which is a histone methyltransferase, and with LSD1, which is a histone demethylase [8]. PUMILIO proteins are members of the PUF family of RNA-binding proteins (RBPs) that exhibit a sequence-specific and highly conserved recognition domain called the PUMILIO Homology Domain (Pum-HD) that interacts with the RNA [9,10]. The interaction occurs at the 3TR end of target RNAs with an 8-nucleotide consensus sequence (UGUANAUA), known as the PUMILIO Response Element (PRE) [11]. PUMILIO proteins repress mRNA by promoting deadenylation of the poly-A tail through the CNOT complex. In addition, the PUMILIO, may also act antagonistically to the PABPC1 protein by repressing translation initiation, preventing the binding of the translation initiation factors eIF4E and eIF5B [11]. Expression of PUM in human tumorigenesis and characterization of PUM-mediated translation control directly in cancer cells has demonstrated its potential to regulate hundreds of transcripts, including many oncogenes and tumor suppressor genes [12].

Long non-coding RNAs (LncRNA) represent a heterogeneous family of RNAs that have emerged as regulators of various biological processes [13]. LncRNAs and RNA-binding proteins (RBP) can associate with ribonucleoprotein (RNP) complexes in both the cytoplasm and the nucleus [14]. They act through distinct mechanisms, controlling gene expression at multiple levels and are involved in physiological and pathological scenarios [15]. Increased expression levels of these LncRNAs can act as RBP decoys, sequestering these proteins and thus hindering their function in regulating their target genes [16]. In this study, we propose a new strategy for searching for potential new biomarkers in breast cancer focused on the regulatory axis formed by LncRNAs, PUMILIO proteins, and their targets in the breast cancer context.

## 2. Results

### 2.1. LncRNA with Sites for PUMILIO Are Differentially Expressed in BRCA

Of the RNAs containing PUMILIO binding sites identified in previous work [17,18], 47 were classified as lncRNAs (Appendix A). From these, by comparing linear models using the Limma package in R, we found 34 as differentially expressed (*p* < 0.05), between tumor and non-tumor samples, in BRCA (Appendix A). We then selected the lncRNAs that showed above the median—log_2_FC cutoff expression in at least one BRCA subtype, resulting in 27 lncRNAs (Figure 1A,B).

Since lncRNAs have intrinsically low expression in human tissue, we selected those that showed detectable expression in at least 80% of the samples from the 1039 patients diagnosed with breast cancer in the Xena Browser platform (https://tcga.xenahubs.net, accessed on accessed on 16 August 2022). Of the twenty-seven differentially expressed lncRNAs with binding sites for PUMILIO, seven showed detectable expression: ASH1L-AS1 (ENSG00000235919), RHPN1-AS1 (ENSG00000254389), SERTAD4-AS1 (ENSG00000203706), NORAD (ENSG00000248360), LINC01011 (ENSG00000244041), LINC00504 (ENSG00000248360), and MALAT1 (ENSG00000251562).

Next, we looked up the relative abundance of the lncRNAs using the same data available in the Xena Browser platform (https://tcga.xenahubs.net, accessed on 16 August 2022). LINC00504 has the highest number of sites for PUMILIO (71). Still, its relative abundance (FPKM = 2.491192) in the samples is much lower when compared to NORAD (FPKM = 8.832009) and MALAT1 (FPKM = 7.336683), which contain seventeen and four PRE (PUMILIO Response Element), respectively. The lncRNAs ASH1L-AS1, RHPN1-AS1, SERTAD4-AS1, and LINC01011 have only one PRE and low expression. Considering that the number of PRE and the relative abundance of each lncRNA can increase or decrease the ability to sequester PUMILIO, we assume that the lncRNAs NORAD, MALAT1, and LINC00504 have a higher potential to influence the regulatory activity of PUMILIO (Figure 1C).

### 2.2. LncRNA NORAD Shows Co-Expression Networks in All Four BRCA Subtypes

For the construction of the co-expression networks, we selected the lncRNAs NORAD, MALAT1, and LINC00504, which showed a higher abundance or more sites for PUMILIO than the other lncRNAs with detectable expression. The analysis did not point to any PUMILIO target genes significantly co-expressed with the MALAT1 or LINC00504 in breast cancer samples. However, we found that PUMILIO targeted genes co-expressed for NORAD in all four subtypes. The *RALGAPB* gene appears co-expressed with NORAD in all four BRCA subtypes and shows the highest mutual information value in the Basal-like subtype (0.55) (*p* < 0.00001) (Figure 2).

The lncRNA NORAD shows negative differential expression (are less expressed) in the Basal-like subtype, so we selected the target genes that were also under-expressed (below the median—log_2_FC cutoff) (*p* < 0.05).

### 2.3. NORAD and RALGAPB Are Differentially Expressed in Lum A and Basal-Like Subtypes Compared to Non-Tumoral Samples

NORAD is differentially expressed and inversely modulated between Lum A and Basal-like. While Lum A has a better prognosis with low-grade tumors, Basal-like is characterized by greater invasive potential [1]. In this context, our goal was to seek explanations that help to understand these differences. Our analyses showed higher expression of NORAD (Log_2_FC = 0.2541, *p* = 2.31 × 10^−5^) and *RALGAPB* (Log_2_FC = 0.3526, *p* = 5.96 × 10^−11^) in Lum A when compared to non-tumor samples and lower expression in Basal-like-NORAD (Log_2_FC = −0.2506, *p* = 4.664 × 10^−4^) and *RALGAPB* (Log_2_FC = −0.1533, *p* = 1.54 × 10^−2^) when compared to non-tumor samples. When comparing Lum A with Basal-like subtypes, both NORAD and *RALGAPB* showed higher expression in Lum A (Log_2_FC = 0.5048, *p* = 1.19 × 10^−16^) (Log_2_FC = 0.5059, *p* = 6.56 × 10^−24^), respectively. In the context of tumor phenotype, NORAD may act differently between the two subtypes and may present a tumor suppressor or oncogenic role [19,20]. Its higher expression in Lum A gives it a greater potential to sequester PUMILIO. Consequently, *RALGAPB*-increased levels may be a consequence of this regulatory axis. The opposite effect could be assumed between NORAD and *RALGAPB* in Basal-like samples (Figure 3A,B).

### 2.4. NORAD and RALGAPB Are up Expressed in Breast Cancer and Related to OS in Patients

We performed Kaplan–Meier and Cox proportional hazards regression analyses to verify if the signatures of NORAD and *RALGAPB* could predict the overall survival (OS) in breast cancer patients.

In Lum A and Basal-like patients, NORAD expression was not found to be related to OS (Figure 4A,B). However, the up expression of *RALGAPB* was associated with a poor OS in Lum A patients (*p* = 0.042) (Figure 4C,D). In Lum A and Basal-like patients, the combined expression of NORAD and *RALGAPB* was also not found to be related to OS (Figure 4E,F).

A univariate Cox regression analysis shows, for both NORAD and *RALGAPB*, an increased expression was related to a worse prognosis and a poor OS in Lum A patients. A multivariate Cox regression analysis confirms these results for NORAD regarding patients’ age (Table 1).

### 2.5. NORAD and RALGAPB Showed a Biomarker Potential, Alone or Combined in Panel

We performed an ROC curve analysis to determine the biomarker potential of NORAD and *RALGAPB*. We found a potential for NORAD to discriminate Basal-like tumor samples (*n* = 190) from non-tumor samples (NT) (*n* = 113) with high sensibility (88.5%) and specificity (72.11%). Despite the limited specificity, the *RALGAPB* gene showed a high sensitivity to identify both Lum A and Basal-like subtypes compared to non-tumor samples, suggesting a modest potential as a biomarker. The combined panel analysis with two molecules increased the sensitivity and specificity of *RALGAPB* to discriminate both Lum A and Basal-like from NT samples. Particularly, NORAD and *RALGAPB* in the panel may distinguish Basal-like from NT samples with 91.15% specificity. Additionally, NORAD and *RALGAPB* combined showed a potential to discriminate Lum A samples from Basal-like (68.95% sensitivity and 76.61% specificity) (Table 2).

## 3. Materials and Methods

### 3.1. TCGA Analysis

The expression data for the lncRNAs were extracted using the GDCRNATools package [21] and the pipeline contained in the Limma package [22] from The Cancer Genome Atlas (TCGA) database through the Xena Browser platform (https://tcga.xenahubs.net accessed on 16 August 2022), following the program’s ethical, legal, and policy criteria. An RNAseq-Illumina HiSeq table was used to extract the differential expression data as log2(FPKM + 1) for lncRNA and mRNA. The clinical data of the patients were extracted using the cBioPortal platform. The BRCA patient data were analyzed into molecular subtypes: Luminal A (Lum A) (*n* = 560), Luminal B (Lum B) (*n* = 207), HER2-enriched (*n* = 82), and Basal-like (*n* = 190) [19].

### 3.2. LncRNAs Selection

The selection of the long non-coding RNAs with PUMILIO biding sites (UGUANAUA) was made using data from [17,18]. Smialek and collaborators [18] used RIP-Seq and RNA-Seq to identify mRNA regulated by PUM1 and PUM2 in the cell line TCam-2, a human male germ cell model. Bohn and collaborators [17] validated a set of RNAs with PUMILIO binding sites in human HEK293 cells using RNA-Seq analysis, statistical testing, and experimental validation. We identified lncRNAs with sites for PUMILIO and with positive expression (above the median—log_2_FC cutoff) (*p* < 0.05), when comparing tumor versus non-tumor tissues in at least one BRCA subtype. Considering the low intrinsic expression of lncRNAs in general, we used one more filter, selecting only lncRNAs expressed in at least 80% of tumor samples.

### 3.3. Co-Expression Regulatory Network

Co-expression networks were generated using the RTN package, which is designed for the reconstruction of transcriptional regulatory networks (TRNs) and analysis of regulons using mutual information (MI) metrics [23]. We tested the association between a given lncRNA and all potential targets using transcriptome data and applied 1000 permutations considering the significance level *p* < 0.00001. The same strategy was previously used to identify potential lncRNA co-expression networks in Lum A breast cancer tumors [21].

Finally, of all the genes selected for each lncRNA with a site for PUMILIO, only those that were also targets of PUMILIO according to [17,18] data and have a positive expression (above the median—log_2_FC cutoff) (*p* < 0.05) when comparing tumor versus non-tumor tissue, were selected. The lncRNA-target gene pairs with co-expression values of >0.15 were chosen, interpreting a correlation above 10% between lncRNA and the target. Each target was analyzed for its expression in tumor tissue compared to normal tissue using only TCGA data.

### 3.4. Statistical Analysis

We analyzed univariate and combined analysis for disease-free (DFS) and overall survival (OS), based on disease-free days and the presence or absence of death events, respectively, for lncRNA and genes co-expressed using the R platform with the survival and survminer packages. We compared the low or high expression of each lncRNA and mRNA targeted by PUMILIO using long-rank (Mantel–Cox tests) and Kaplan–Meier curves. Patient survival was analyzed according to each gene expression and to the pairwise expression in each patient. A *p* < 0.05 was considered statistically significant.

To verify whether the signature of the two genes, NORAD and *RALGAPB*, can predict the survival of breast cancer patients independently of age, we applied multivariate Cox regression analyses.

ROC (receiver operating characteristic) curves were performed using 1039 patients (Lum A *n* = 560, Lum B *n*= 207, Her2-enriched *n* = 82, and Basal-like *n* = 190) and 113 non-tumor samples. Combined ROC curves were performed through binary logistic regression analysis using IBM SPSS Statistics 26.0 (IBM SPSS Statistics Inc., Armonk, NY, USA). Differential expression data of lncRNA and PUMILIO genes of the BRCA cohort were used.

The cutoff values of the ROC curves, sensitivity, and specificity were determined by the Youden index. Analyses were performed using GraphPad Prism v.9.0.0 (GraphPad Software lnc., San Diego, CA, USA).

## 4. Discussion

From the RNAs containing PUMILIO binding sites identified in previous work [17,18], we identify 27 lncRNA with sites for PUMILIO positively regulated (above the median—log_2_FC cutoff) (*p* < 0.05) in the BRCA samples. Of these, the NORAD lncRNA showed the highest relative abundance in the cellular environment (FPKM = 8.832009) and appeared co-expressed with the *RALGAPB* gene in all four breast cancer subtypes: Lum A, Lum B, Her2-enriched, and Basal-like, with the highest mutual information value in the Basal-like subtype (0.55) (*p* < 0.00001).

Taking into account the heterogeneity among BRCA subtypes, in other words, between Lum A and Basal-like subtypes, Lum A has a better prognosis, characterized by low-grade tumors, while Basal-like has a worse prognosis and is characterized by tumors with greater invasive potential [1]. Thus, we verified the expression of both genes in these subtypes.

Our analyses showed higher expression of NORAD (Log_2_FC = 0.2541, *p* = 2.31 × 10^−5^) and *RALGAPB* (Log_2_FC = 0.3526, *p* = 5.96 × 10^−11^) in Lum A when compared to non-tumor samples and lower expression in Basal-like-NORAD (Log_2_FC = −0.2506, *p* = 4.664 × 10^−4^) and *RALGAPB* (Log_2_FC = −0.1533, *p* = 1.54 × 10^−2^) when compared to non-tumor samples. When comparing Lum A with Basal-like subtypes, both NORAD and *RALGAPB* showed higher expression in Lum A (Log_2_FC = 0.5048, *p* = 1.19 × 10^−16^; Log_2_FC = 0.5059, *p* = 6.56 × 10^−24^, respectively). In the context of tumor phenotypes, NORAD may be acting in different ways between the two subtypes and may present a tumor suppressor or oncogenic role [19,24].

Our work suggests that the higher expression of NORAD in Lum A gives it a higher potential to sequester PUMILIO. Consequently, the increased levels of *RALGAPB* may be a consequence of this regulatory axis. The opposite effect could be assumed between NORAD and *RALGAPB* in Basal-like.

PUMILIO (PUM) proteins are members of an evolutionarily conserved family of RBPs known as PUF (Pumilio-Fem-3-binding factor) [25]. The characterizing feature of all PUF proteins is the presence of the C-terminal RNA-binding domain known as the PUMILIO homology domain (PUM-HD), formed by eight imperfect tandem repeats of 36 amino acids each [12]. PUMILIO acts in the repression of translation of messenger RNAs with which it interacts through domains in its structure capable of recognizing specific nucleotide sequences in the 3’UTR region of the target transcript. Translation repression can occur through the recruitment of factors that will prevent translation or through the degradation of the target mRNA by the recruitment of deadenylase complexes [25]. PUMILIO is also an important regulator of DICER1, indirectly affecting miRNA processing in DICER1 syndrome, a cancer predisposition disorder [26].

Loss of PUMILIO-mediated fine regulation could provide ideal conditions for tumor development, as cancer cells would benefit from the proteomic plasticity generated to meet their demands for growth and metastasis [27].

NORAD is highly conserved and an abundant lncRNA, with expression levels of approximately 500 to 1000 copies per cell. This lncRNA maintains genomic stability by sequestering PUMILIO proteins, preventing them from repressing target mRNA stability and translation. In the absence of NORAD, the hyperactivity of PUMILIO represses mitotic, DNA repair, and replication factors [13].

Similar to some other lncRNAs, such as XIST and FIRRE [28], the NORAD sequence contains repeat elements—12 NORAD repeat units (NRUs)—recognizable and sequence-like, which likely originated by tandem duplication in mammalian emergence and still share substantial sequence homology. Most NRUs contain one or two binding sites for the two homologs of PUMILIO in mammals: PUM1 and PUM2. NORAD contains at least 17 of these PUM recognition elements (PREs), which have the consensus sequence UGURNAUA (R = A/G). This structural domain organization facilitates its function on multiple levels [29]. NORAD interacts with the PUMILIO proteins (PUM1/PUM2), controlling their availability in the cytoplasm. Recently, it was shown that NORAD promotes the condensation and sequestration of PUM proteins into NP (NORAD–PUMILIO) bodies [29].

Deletion of NORAD in cell lines results in increased repression of genes carrying PREs in their 3’UTR regions, and overexpression has an inverse effect [13]. Since these proteins regulate a wide class of transcripts and many of them are involved with genome protection and cell cycle control, changes in NORAD levels could consequently unbalance the orchestration of essential mechanisms for cellular homeostasis [30].

We know that NORAD-regulated PUM targets are enriched in genes involved in cell division [30], and depletion of this lncRNA results in defects in mitosis and the accumulation of cells with instability in chromosome number [13].

Our analyses showed the *RALGAPB* gene co-expressed with NORAD in all four BRCA subtypes with the highest mutual information value in the Basal-like subtype (0.55) (*p* < 0.00001).

RalGAPs (Ral GTPase activating proteins) are multi-subunit complexes comprising a catalytic alpha (α) subunit and beta regulatory (β) subunits. They act as GTPase activators for the small GTPases RALA and RALB. The function and stability of the complex require the association of the α and β subunits.

During mitosis, *RALGAPB* shows a dramatic cell redistribution, shifting from the nucleus and Golgi complex during the prophase to the mitotic spindle and intercellular bridge at cytokinesis [31]. Since the *RALGAPB* is a target of PUMILIO, increased or decreased expression of lncRNA NORAD interferes with the efficiency of interaction with PUMILIO, and should lead to increased or decreased *RALGAPB* gene expression in BRCA.

A study evaluating the role of *RALGAPB* during the process of cell mitosis showed that its depletion causes chromosomal misalignment and decreases the amount of mitotic cyclin B1, disturbing the transition from metaphase to anaphase. The over-expression of *RALGAPB* interferes with cell division, leading to binucleation and multinucleation, and cell death [31].

A study in oral squamous cell carcinoma [10] showed that *RALGAPB* silencing increased the level of RalA activation, promoting HSC-2 cell migration and invasion in vitro, a profile characteristic of metastatic cells. In HeLa cells, the depletion of the *RALGAPB* protein in the mitotic spindle and intercellular bridge caused a considerable increase in the number of metaphase cells with aberrant chromosomal misalignment in the equatorial plane [31].

Applying Kaplan–Meier analyses, we found that up expression of *RALGAPB* was related to the poor OS in patients with Lum A breast cancer (*p* = 0.042). The signature of this gene and the NORAD can predict a poor prognosis in Lum A. The results of the univariate and multivariate Cox analysis for NORAD and *RALGAPB* again point out that the higher expression of NORAD is related to a worse prognosis for Lum A patients compared to the *RALGAPB* gene. Furthermore, we also verified the independence of these results considering the patients’ age. The multivariate Cox analysis showed that the poor OS found in Lum A patients regarding NORAD expression is independent of age. Moreover, the multivariate analyses also do not confirm the independence of the two genes in predicting prognosis, which may support our hypothesis that a NORAD–PUMILIO–*RALGAPB* regulatory axis is involved with the tumor phenotype and is impacting patient survival.

Finally, the results of the ROC curve analyses showed that the differential expression of NORAD has an excellent potential to discriminate Basal-like tumor samples (*n* = 190) from non-tumor samples (*n* = 113) (88.5% sensitivity, 72.11% specificity, AUC = 0.8496, CI: 0.80–0.89, *p* < 0.001), suggesting its potential as a biomarker in Basal-like patients. Additionally, the combined panel with NORAD and *RALGAPB* increased these molecules’ biomarker potential, especially to distinguish Basal-like from NT samples. Nonetheless, the potential to discriminate Lum A and Basal-like in the panel is also achievable.

The signature of NORAD and *RALGAPB* in Lum A and Basal-like subtypes can predict a worse or better prognosis according to the survival of patients, and also has the potential to discriminate tumor and non-tumor groups. Functional analyses must be conducted to validate our findings.

## 5. Conclusions

In this study, we propose a new strategy for searching for potential new biomarkers in breast cancer focused on the regulatory axis formed by LncRNAs, PUMILIO proteins, and their targets in the breast cancer context. Our results pointed out NORAD as the most relevant LncRNA with PUMILIO binding site in breast cancer, differently expressed between Lum A and Basal-like subtypes and co-expressed significantly in Basal-like (0.55) with the *RALGAPB* gene, a target gene of PUMILIO, related to chromosome stability during cell division. This work makes *RALGABP* a new player in genomic instability promoted by NORAD dysregulation in breast cancer.

## Figures and Tables

**Figure 1 ncrna-08-00076-f001:**
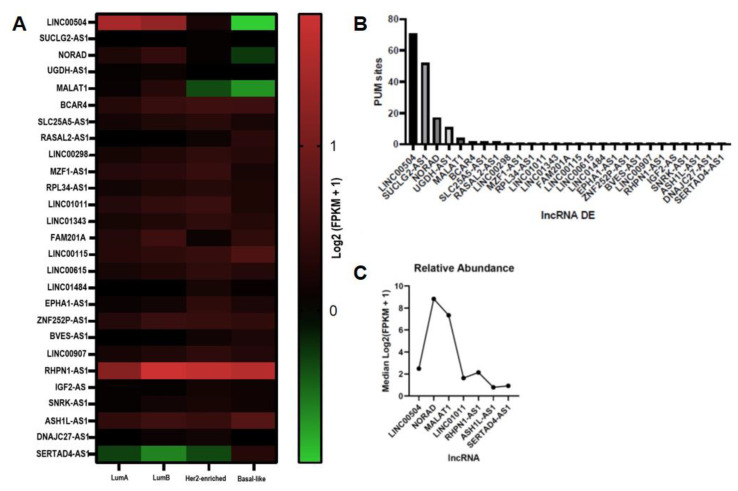
LncRNA with sites for PUMILIO differentially expressed in BRCA. Legend: (**A**) Heatmap representation of the activity of expression networks of 27 lncRNAs in breast cancer molecular subtypes. According to this metric, differentially expressed positive (above the median—log2FC cutoff) (*p* < 0.05) are in red and differentially expressed negative (below the median—log2FC cutoff) (*p* < 0.05) are in green. Samples are arranged in the columns of the heatmap and categorized according to the annotation bar on the side of the graph (−0.92 to 1.8). The TCGA BRCA cohort samples were classified according to molecular classification into Lum A (*n* = 560), Lum B (*n* = 207), Her2-enriched (*n* = 82), and Basal-like (*n* = 190). (**B**) Number of sites for PUMILIO in 27 differentially expressed lncRNAs in breast cancer (BRCA). (**C**). Relative abundance of seven lncRNA with detectable expression—median Log2(FPKM + 1) (Fragments Per Kilobase Million) in BRCA samples from the same TCGA BRCA cohort.

**Figure 2 ncrna-08-00076-f002:**
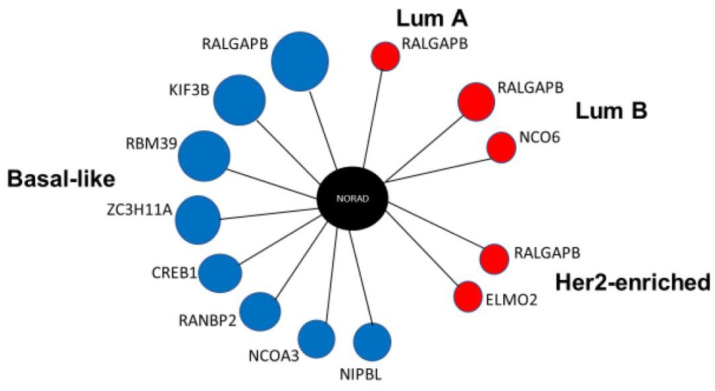
Co-expression network for NORAD lncRNA in BRCA subtypes. Legend: Network representation of NORAD extended in genes with higher values of mutual information in BRCA subtypes. Red circles refer to genes with high expression and blue to low expression compared to non-tumor samples in each BRCA subtype and all show a direct correlation with NORAD. The size of the dot refers to the mutual information value, whereas large dots indicate greater mutual information.

**Figure 3 ncrna-08-00076-f003:**
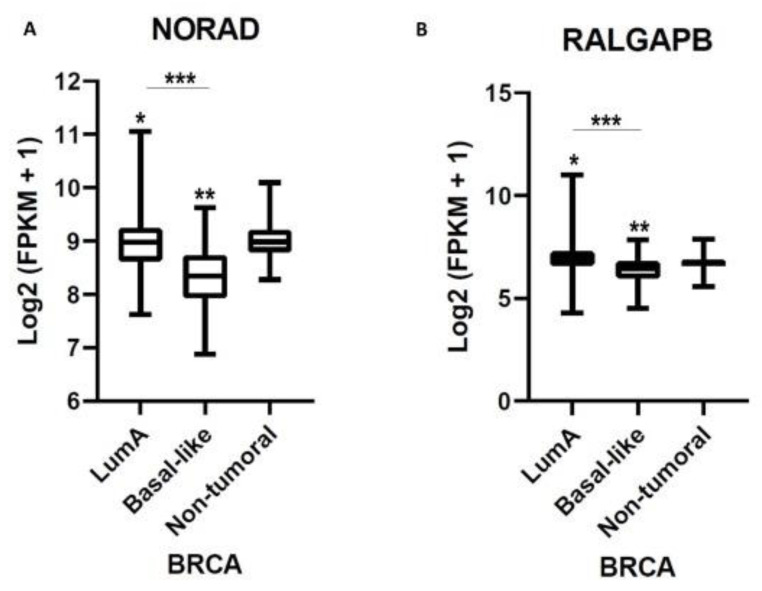
Differential expression of NORAD lncRNA and *RALGAPB* for Lum A and Basal-like subtypes. Legend: (**A**) Differential expression of NORAD lncRNA in comparing Lum A, Basal-like subtypes, and non-tumor samples. (NORAD) Lum A—Log_2_FC = 0.254, * *p* = 1.19 × 10^−16^, Basal-like—Log_2_FC = −0.2506, ** *p* = 4.66 × 10^−4^, Lum A versus Basal-like—Log_2_FC = 0.5048, *** *p* = 1.19 × 10^−16^. (**B**) Differential expression of *RALGAPB* in comparing Lum A, Basal-like subtypes, and non-tumor samples. (*RALGAPB*) Lum A—Log_2_FC = 0.3526, * *p* = 5.96 × 10^−11^, Basal-like—Log2FC = −0.1533, ** *p* = 1.542 × 10^−2^, Lum A versus Basal-like—Log2FC = 0.5059, *** *p* = 6.56 × 10^−24^.

**Figure 4 ncrna-08-00076-f004:**
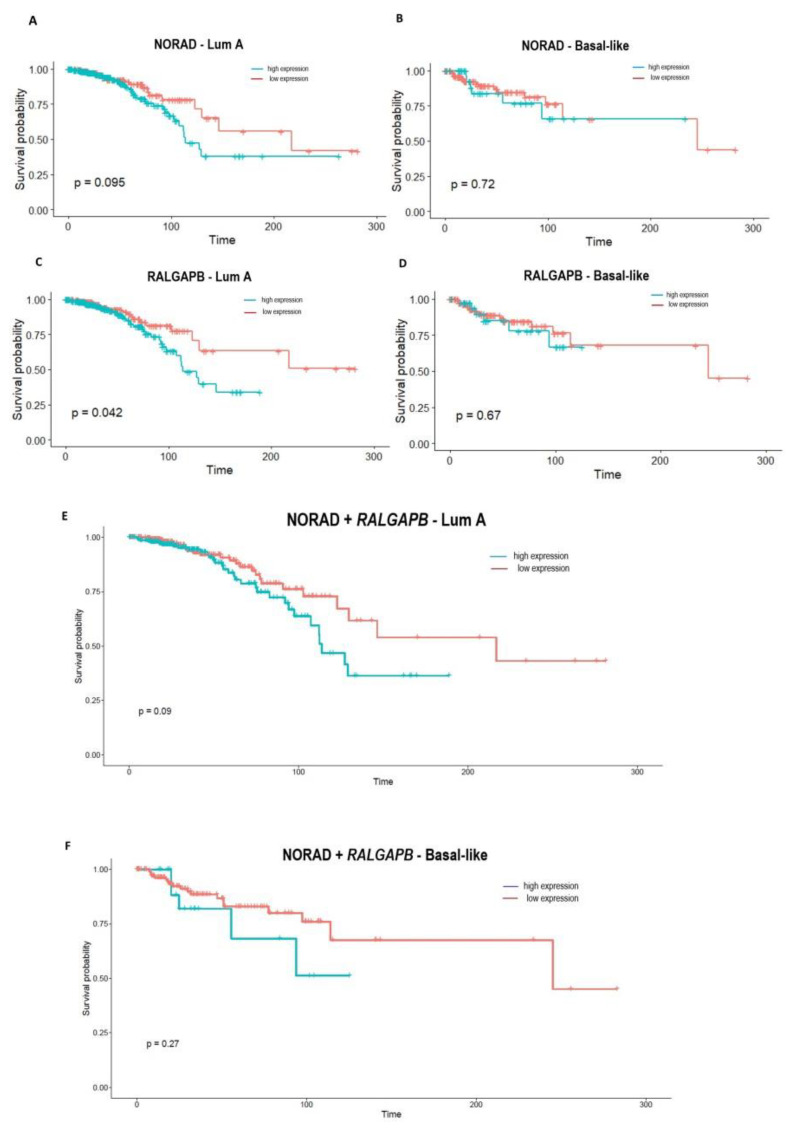
Overall survival analyses for Lum A and Basal-like patients related to lncRNA NORAD and *RALGAPB* expression. **Legend:** (**A**) Survival analysis considering the median value of NORAD in Lum A (High expression (*n* = 304), Low expression (*n* = 185)) and (**B**) Basal-like patients (High expression (*n* = 37), Low expression (*n* = 131)). (**C**) Survival analysis considering the median value of *RALGAPB* in Lum A (High expression (*n* = 282), Low expression (*n* = 207)), and (**D**) Basal-like patients (High expression (*n* = 38), Low expression (*n* = 130)). (**E**) Survival combined panel analysis with NORAD and *RALGAPB* in Lum A patients. (**F**) Survival combined panel analysis with NORAD and *RALGAPB* in Basal-like patients. Red and blue lines refer to patients with low and high expression, respectively. A *p*-value < 0.05 was considered significant.

**Table 1 ncrna-08-00076-t001:** Univariate and Multivariate Cox regression analyses for NORAD and *RALGAPB* in Lum A patients.

Variables Lum A	Univariate Model			Variables Lum A	Multivariate Model		
	HR	CI	*p*-value		HR	CI	*p*-value
NORAD	2	1.2–3.4	0.01	NORAD	2.73	1.07–6.9	0.035
*RALGAPB*	1.6	0.95–2.8	0.079	*RALGAPB*	0.72	0.29–1.8	0.475
				AGE	1.04	1.02–1.1	<0.001

Legend: HR: Hazard Ratio; CI 95%: Confidence Interval of 95%.

**Table 2 ncrna-08-00076-t002:** Data about receiver operating characteristic (ROC) curves to investigate the diagnostic potential of RNAs in Lum A and Basal-like samples.

Comparison	RNA	AUC	Sensitivity	Specificity	*p*-Value
*LA* × *NT*	NORAD	0.5369	86.73	29.64	=0.2149
	*RALGAPB*	0.6574	88.50	46.96	<0.0001
	NORAD + *RALGAPB*	0.7242	78.79	56.79	<0.0001
*BL* × *NT*	NORAD	0.8496	88.50	72.11	<0.0001
	*RALGAPB*	0.6659	85.84	51.58	<0.0001
	NORAD + *RALGAPB*	0.8588	70.53	91.15	<0.0001
*LA* × *BL*	NORAD	0.7953	69.47	76.07	<0.0001
	*RALGAPB*	0.7555	81.58	57.86	<0.0001
	NORAD + *RALGAPB*	0.7980	68.95	76.61	<0.0001

Legend: Sensitivity and specificity are presented as a percentage (%). In bold and underlined are the highest AUC value for the group comparison. NT = non-tumor; LA = luminal A; BL = basal-like.

## Data Availability

Not applicable.

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
