# Peer review of "The Potential of NORAD–PUMILIO–RALGAPB Regulatory Axis as a Biomarker in Breast Cancer"

_ncrna, 2022, doi:10.3390/ncrna8060076_

Round 1
Reviewer 1 Report
In this study, authors mined in BRCA TCGA datasets to identify lncRNAs with PUMILIO binding sites and additionally stratified the samples based on their molecular subtypes into Luminal A, B basal and HER2 enriched. Upon insilco analysis, authors identified a regulatory axis consisting of a lncRNA NORAD with gene RALGABP and proposed that this axis due to the capability to sequester PUMILIO proteins can predict worse or better prognosis in breast cancer based on the available survival data. Authors further highlight that these findings need to be experimentally validated.
While this study attempts to answer a critical question when it comes to predicting the prognosis in breast cancer, it would be more scientifically sound if following points are addressed.
1. Authors only consider the molecular subtypes of breast cancer (Luminal A, Luminal B, Basal and HER2 enriched), authors should consider stratifying the BRCA TCGA dataset into histological subtypes of breast cancer such as ductal and lobular breast cancer.
2. LncRNAs are a heterogenous group of molecules which interact with DNA, RNA as well as proteins. While ribonucleoprotein complexes are at the heart of its functions, authors should consider constructing the same signature using additional RNA binding proteins such as and not limited to HNRNPs/ YBX1/ELAV1 (all commonly found associated with lncRNAs) to validate their pipeline for predicting the prognosis in BC.
Author Response
In this study, authors mined in BRCA TCGA datasets to identify lncRNAs with PUMILIO binding sites and additionally stratified the samples based on their molecular subtypes into Luminal A, B basal and HER2 enriched. Upon in silico analysis, authors identified a regulatory axis consisting of a lncRNA NORAD with gene RALGABP and proposed that this axis due to the capability to sequester PUMILIO proteins can predict worse or better prognosis in breast cancer based on the available survival data. Authors further highlight that these findings need to be experimentally validated.
While this study attempts to answer a critical question when it comes to predicting the prognosis in breast cancer, it would be more scientifically sound if following points are addressed.
- Authors only consider the molecular subtypes of breast cancer (Luminal A, Luminal B, Basal and HER2 enriched), authors should consider stratifying the BRCA TCGA dataset into histological subtypes of breast cancer such as ductal and lobular breast cancer.
Thanks for your suggestion. In fact, we considered the stratification of BRCA samples into ductal and lobular important, but the number of lobular samples among Lum B (n=11), Her2-enriched (n=2) and Basal-like (n=2) subtypes is very low, not allowing a statistically significant evaluation.
- LncRNAs are a heterogenous group of molecules which interact with DNA, RNA as well as proteins. While ribonucleoprotein complexes are at the heart of its functions, authors should consider constructing the same signature using additional RNA binding proteins such as and not limited to HNRNPs/ YBX1/ELAV1 (all commonly found associated with lncRNAs) to validate their pipeline for predicting the prognosis in BC.
We agree that investigating the modulation of other RBPs is extremely relevant. We add in the article’s introduction a paragraph commenting about the role of RBPs and lncRNAs regulating gene expression (lines 74 – 81).
In this work we focus on lncRNA NORAD and the PUMILIO protein, having as main objective the evaluation of this regulatory axis. We also considered the fact that the relationship between lncRNA NORAD and PUMILIO is well established, which encouraged our work. Although this paper focuses on PUMILIO, we understand the importance of evaluating other RBPs in the tumor context and will consider these points in our next work.
Reviewer 2 Report
Muller and colleagues applied bioinformatics tools to investigate long-noncoding RNAs (lncRNAs) associating with PUMILIO RNA-binding protein (RBP) in a breast cancer model, utilizing publicly available RNA sequencing databases. They found that lncRNA, NORAD has PUMILIO binding sites, is differently expressed in four subtypes of breast cancer, and co-expresses with PUMILIO target gene RALGAPB.
Given a poorly understood yet regulatory function of lncRNA molecules, the findings are interesting and add an additional piece of information for the role of lncRNAs in cancer cell abnormality.
To improve the manuscript, the authors need to address the specific comments as follows:
1. After the initial exploration’s results (Fig.1), the authors excluded from further analysis lncRNA LINC00504 stating that it has lower expression in breast cancer samples. Even if LINC00504 is expressed at lower levels it still could have regulatory value, since it has a high number of PUMILIO binding sites (as showed in Fig 1B) and is differentially expressed in breast cancer subtypes (Fig. 1A). Therefore, the authors should also include LINC00504 to co-expression network analysis, as they did for NORD and MALAT1, and address the results in the manuscript.
2. The authors found the PUMILIO target gene, RALGAPB is co-expressed with NORAD lncRNA in all four subtypes: Luminal A, Luminal B, Her2-enriched, and Basal-like (Fig. 2). However, in the subsequent part of the manuscript they focus only on two subtypes, Lum A and Basal-like (Fig. 3 and 4). The authors should either add subtypes Lum B and Her2-enriched to the results in Fig. 3 and 4, or clearly explain the reasoning for omitting these two subtypes from downstream analysis.
Author Response
Muller and colleagues applied bioinformatics tools to investigate long-noncoding RNAs (lncRNAs) associating with PUMILIO RNA-binding protein (RBP) in a breast cancer model, utilizing publicly available RNA sequencing databases. They found that lncRNA, NORAD has PUMILIO binding sites, is differently expressed in four subtypes of breast cancer, and co-expresses with PUMILIO target gene RALGAPB.
Given a poorly understood yet regulatory function of lncRNA molecules, the findings are interesting and add an additional piece of information for the role of lncRNAs in cancer cell abnormality.
To improve the manuscript, the authors need to address the specific comments as follows:
- After the initial exploration’s results (Fig.1), the authors excluded from further analysis lncRNA LINC00504 stating that it has lower expression in breast cancer samples. Even if LINC00504 is expressed at lower levels it still could have regulatory value, since it has a high number of PUMILIO binding sites (as showed in Fig 1B) and is differentially expressed in breast cancer subtypes (Fig. 1A). Therefore, the authors should also include LINC00504 to co-expression network analysis, as they did for NORAD and MALAT1, and address the results in the manuscript.
Thanks for your considerations. The LINC00504 was included in the co-expression analyses, but we didn´t find any significant correlation with PUMILIO target genes. We assumed that due its low expression, the impact on sequestering PUMILIO didn´t reach the significancy threshold on affecting PUM targets. We understand that this was not clear, and we will include it in the body of the article ( lines 196, 202, 204).
- The authors found the PUMILIO target gene, RALGAPB is co-expressed with NORAD lncRNA in all four subtypes: Luminal A, Luminal B, Her2-enriched, and Basal-like (Fig. 2). However, in the subsequent part of the manuscript they focus only on two subtypes, Lum A and Basal-like (Fig. 3 and 4). The authors should either add subtypes Lum B and Her2-enriched to the results in Fig. 3 and 4, or clearly explain the reasoning for omitting these two subtypes from downstream analysis.
We agree with the reviewer that its important to look for all BC molecular subtypes in these analysis. In fact, we consider the investigation of the other subtypes important, however, since lncRNA NORAD is differentially expressed and inversely modulated between Lum A and Basal-like tumors, we focused on these groups. Besides, as Luminal A patients show low-grade tumors and good prognosis and Basal-like subtypes comprise patients with worse prognosis and higher-grade tumors (Naito & Urasaki, 2018), our goal was to seek explanations that help to understand these differences. We agree that this information is unclear in the text and we will add it on the article ( lines 222, 224, 225 ).
Round 2
Reviewer 1 Report
Authors have answered satisfactorily to the queries